# Selenium Nanoparticles Suppressed Oxidative Stress and Promoted Tenocyte Marker Expression in Tendon-Derived Stem/Progenitor Cells

**DOI:** 10.3390/antiox13121536

**Published:** 2024-12-15

**Authors:** Pauline Po Yee Lui, Caihao Huang, Xing Zhang

**Affiliations:** 1Department of Orthopaedics and Traumatology, The Chinese University of Hong Kong, Shatin, New Territories, Hong Kong SAR, China; 2Center for Neuromusculoskeletal Restorative Medicine Ltd., Hong Kong Science Park, Shatin, New Territories, Hong Kong SAR, China; 3Institute of Metal Research, Chinese Academy of Sciences, Shenyang 110000, China; caihao@mail.ustc.edu.cn (C.H.); xingzhang@imr.ac.cn (X.Z.); 4School of Materials Science and Engineering, Dalian University of Technology, Dalian 116000, China

**Keywords:** inflammation, oxidative stress, selenium, tendon-derived stem/progenitor cells (TDSCs)

## Abstract

Traumatic tendon injuries generate reactive oxygen species and inflammation, which may account for slow or poor healing outcomes. Selenium is an essential trace element presented in selenoproteins, many of which are strong antioxidant enzymes. Selenium nanoparticles (SeNPs) have been reported to promote tissue repair due to their anti-oxidative, anti-inflammatory, anti-apoptotic, and differentiation-modulating properties. However, its effects on the functions of tendon-derived stem/progenitor cells (TDSCs) and tendon healing have not been reported. This study examined the effects of SeNPs on the functions of hydroperoxide (H_2_O_2_)-stimulated TDSCs. Rat patellar TDSCs were treated with H_2_O_2_ with or without SeNPs. The viability, marker of proliferation, oxidative stress, inflammation, apoptosis, and tenocyte marker expressions of H_2_O_2_-stimulated TDSCs after SeNPs treatment were assessed. Our results showed that SeNPs increased the viability and expression of the marker of proliferation of TDSCs exposed to H_2_O_2_, while concurrently reducing oxidative stress, inflammation, and apoptosis. Additionally, the expressions of tenocyte markers were significantly elevated in H_2_O_2_-treated TDSCs after treatment with SeNPs. Furthermore, the expressions of *Sirt1* and *Nrf2* also increased after SeNPs treatment in H_2_O_2_-stimulated TDSCs. In conclusion, SeNPs mitigated oxidative stress, inflammation, and apoptosis while enhancing the survival and expression of the marker of proliferation of TDSCs in an oxidative stress environment. Additionally, it promoted the fate of TDSCs towards the tenocyte lineage in the presence of such oxidative stress. The increased expressions of *Sirt1* and *Nrf2* likely mediated the anti-oxidative and anti-inflammatory effects of SeNPs. SeNPs hold promise as a novel intervention for promoting tendon healing.

## 1. Introduction

Tendons and ligaments endure substantial tensile stresses and are prone to rupture due to overuse or trauma, resulting in significant pain and functional impairment. The regenerative capacity of tendons and ligaments following an injury is generally poor, which represents a significant burden to the healthcare system. Both conservative treatments and surgical interventions for tendon injuries often yield unsatisfactory outcomes characterized by extended healing time, formation of scar tissue, adhesions, occasional bone formation, and a higher likelihood of re-injury. The accumulation of microinjuries during loading as a result of ineffective tendon regeneration after acute injuries is a predisposing factor for the development of chronic tendinopathy, presented with persistent pain and functional disability over the long term.

Following exercise and traumatic injury, tendons constantly produce reactive oxygen species (ROS) [1,2]. Excessive ROS exceeding the tendon’s antioxidant capacity can trigger inflammation, causing tendon injury. The accumulation of microinjuries may lead to tissue fibrosis and impaired healing [3]. Notably, elite professional soccer players exhibiting ultrasonographic (US) evidence of tendon damage showed significantly higher levels of oxidative stress compared to those with normal US findings [4]. The level of ROS was also elevated in the injured tendon compared to the normal tendon in a patellar partial transection rat model [5]. Previous studies have shown that the tenogenic properties of tendon-derived stem/progenitor cells (TDSCs) were compromised, with increased osteo-chondrogenic differentiation potential in an inflammatory environment [6,7,8]. Therefore, more regenerative healing is expected through reducing oxidative stress and inflammation.

Selenium (Se) is an essential trace element in human health and diseases including immune response, neurodegeneration, cardiovascular diseases, and cancer [9,10,11,12]. The protective effect of selenium on mammalian cells is mainly carried out by a series of selenoproteins in which selenium is incorporated as selenocysteine. There are 25 selenoproteins in the human body and most of them are strong antioxidant enzymes [13]. Pretreatment of human dermal fibroblasts with selenium nanoparticles (SeNPs) reduced cell death induced by heat shock [14]. In addition, SeNPs also inhibited prostaglandin E2 (PGE_2_), which is an activator of pro-inflammatory cytokines like interleukin-6 (IL-6) and tissue necrosis factor alpha (TNF-α) [15]. In addition, SeNPs reduced apoptosis and intracellular ROS levels as well as accelerated myogenic differentiation of C2C12 in the presence of H_2_O_2_ [16]. The incorporation of SeNPs in the wound dressing made of bacterial cellulose/gelatin enhanced the anti-inflammatory, anti-oxidative, and anti-bacterial effects of the hydrogel [17]. However, there has been no study about the effects of SeNPs on TDSCs and tendon healing.

This study aimed to examine the effects of SeNPs on the functions of H_2_O_2_-stimulated TDSCs. We hypothesized that SeNPs would increase the viability, proliferation, and tenogenesis while abrogating oxidative stress, inflammation and apoptosis of TDSCs under oxidative stress. The effects of SeNPs on the expressions of *Sirt1*, *FoxO1*, and *Nrf2*, key regulators of oxidative stress and inflammation, were also investigated.

## 2. Materials and Methods

### 2.1. Study Design

The study was approved by both the animal research ethics committee (Ref. 23-229-NSF, approved on 5 January 2024) and the University Laboratory Safety Office. Adherence to the ARRIVE guidelines was ensured throughout the research process.

TDSCs were seeded in a culture plate in complete culture medium (α-Modified Eagle’s Medium (α-MEM) containing 10% fetal bovine serum (FBS), 100 U/mL penicillin, 100 μg/mL streptomycin; all from Invitrogen Corporation, Waltham, MA, USA) overnight. The cells were washed with phosphate-buffered saline (PBS) and treated with different concentrations of SeNPs (0–2 µg/mL) and H_2_O_2_ (200 µM) in a serum-free medium. At various times after treatment, the viability, marker of proliferation, oxidative stress, inflammation, apoptosis, expressions of tenocyte markers, and transcription factors regulating oxidative defense mechanisms and inflammation of TDSCs were assessed. Hence, H_2_O_2_ was present during the whole experiment. The viability and expression of the marker of proliferation of TDSCs were assessed by AlamarBlue assay and immunofluorescence (IF) staining of Ki67, respectively. The oxidative stress of TDSCs was assessed by CM-H_2_DCF-DA assay kit, mRNA expressions of glutathione peroxidase 1 (*Gpx1*), glutathione peroxidase 3 (*Gpx3*), glutathione peroxidase 4 (*Gpx4*), thioredoxin reductases 2 (*Txnrd2*), selenoprotein M (*Selenom*), superoxidase dismutase (*Sod1*), catalase (*Cat*), NADPH oxidase 1 (*Nox1*), heme oxygenase 1 (*Hmox1*), IF staining of nitrotyrosine and malonaldehyde (MDA), and catalase activity. The catalase activity of TDSCs was measured using an assay kit. The mRNA (*Il6*, *Il1b*, *Cox2*) and protein (IL-6, Cox-2) expressions of inflammatory cytokines were measured by qRT-PCR and IF, respectively. Apoptosis of TDSCs was examined by mRNA expressions of pro-apoptotic (*Bax*, *Bad*, *Bid*, *Casp3*) and anti-apoptotic markers (*Bcl2l1*, *Bcl2*), and IF staining of Bax. The mRNA expressions of tenocyte markers (*Col1a1*, *Col3a1*, *Eln*, *Tnc*, *Dcn*) and transcription factors regulating oxidative defense mechanisms and inflammation (*Sirt1*, *FoxO1*, *Nrf2*) of TDSCs were examined by qRT-PCR.

### 2.2. Preparation of SeNPs

SeNPs were prepared in our laboratory and the quality including the crystallinity, shape, purity, hydrated particle size, and zeta potential were examined. Selenous acid (13 mM) was dissolved in 5 mg/mL of polyvinyl alcohol (PVA) and then mixed with ascorbic acid (33 mM) with vigorous stirring. The selenium particle size was controlled by adjusting the concentration of reactants and reaction times in the reduction reaction of ascorbic acid. The crystallographic structure of the selenium particles was examined by X-ray diffraction analysis (XRD, Rigaku D/max 2400 diffractometer, Rigaku Corporation, Tokyo, Japan) using a monochromated Cu Kα radiation source with an accelerating voltage of 40 kV and a current of 250 mA. The elemental composition of the selenium particles was analyzed by energy dispersive X-ray spectrometer (EDX, QUANTA 450, FEI, Hillsboro, OR, USA) coupled with transmission electron microscopy (TEM, Tecnai F30, FEI, Hillsboro, OR, USA). The zeta potential and hydrated particle size of the selenium particles were measured by a zeta potential analyzer and laser particle size analyzer (ZEN3690, Malvern Panalytical, Malvern, UK), respectively. SeNPs were dissolved in deionized water at 6 mg/mL and sonicated for 30 min. The solution was further diluted with a cell culture medium before use.

### 2.3. TDSC Isolation

Patellar TDSCs were isolated from male Sprague Dawley rats (6–8 weeks, 150–220 g). The procedures for TDSC isolation are well-established [18]. Briefly, the tendon tissue was minced, and enzymatically digested using type I collagenase (Cat no.: C0130, Sigma Aldrich, St. Louis, MO, USA) to produce a single cell suspension, seeded at an appropriate low cell density to isolate the stem cells, and then cultured the cells to facilitate colony formation. The expressions of stem cell markers, clonogenicity, and multi-lineage differentiation potential of the isolated cells were determined according to our established protocols [18]. Only TDSCs at passages fewer than 5 were used for experiments.

### 2.4. Alamar Blue Assay

TDSCs at 4000 cells were seeded in a 96-well plate and treated with or without different concentrations of SeNPs (0–2 µg/mL) and H_2_O_2_ (200 µM) (Cat no.: 23E124006, VWR, Philadelphia, PA, USA) in serum-free medium. At 1 h and 4 h after treatment, the AlamarBlue solution (Cat no.: DAL1100, Thermo Fisher Scientific, Waltham, MA, USA) was added for 2 h. The fluorescence intensity was measured at the excitation wavelength of 560 nm and the emission wavelength of 590 nm. Cell viability relative to the untreated group was calculated and compared among different groups. There were 5 samples/group.

### 2.5. Immunofluorescence Staining of Ki67, IL-6, Cyclooxygenase-2 (Cox-2), Bax, Nitrotyrosine, and Malonaldehyde (MDA)

TDSCs were seeded on slides and treated with 0.5 µg/mL of SeNPs and H_2_O_2_ (200 µM) in a serum-free medium. At 4 h after H_2_O_2_ exposure, the cells were washed, fixed with 4% paraformaldehyde (Cat no.: A11313.36, Thermo Fisher Scientific, Waltham, MA, USA), and permeabilized with 0.3% Triton-X100 (Cat no.: 17-1315-01, Plusone^TM^, GE Healthcare Bio-Sciences KK, Tokyo, Japan). The primary antibodies against Ki67 (Cat no.: 11-5698-82, Thermo Fisher Scientific, Waltham, MA, USA, 1:1000 dilution), IL-6 (Cat no.: ab290735, Abcam, Cambridge, UK, 1:1000 dilution), Cox-2 (Cat no.: A1253, Abclonal Technology, Woburn, MA, USA, 1:1000 dilution), Bax (Cat no.: #MA5-32031, Thermo Fisher Scientific, Waltham, MA, USA, 1:1000 dilution), nitrotyrosine (Cat no.: 9691S, Cell Signal Technology, Danvers, MA, USA, 1:1000 dilution), or MDA (Cat no.: ab27642, Abcam, Cambridge, UK, 1:1000 dilution) were added for 1 h at room temperature. After washing, the cells were treated with goat anti-rabbit IgG (H + L) secondary antibodies conjugated with Alexa FluorTM 488 (Cat no.: A-11008, Thermo Fisher Scientific, Waltham, MA, USA, 1:500 dilution) for 45 min at room temperature. The slides were counter stained with 4′,6-diamidino-2-phenylindole (DAPI) (Cat no.: P36935, Invitrogen Corporation, Waltham, MA, USA). The fluorescence signals were viewed using a fluorescent microscope (Leica DM5500 B, Leica Microsystems, Wetzlar, Germany). The percentages of Ki67-positive cells (3 samples/group), IL-6-positive cells (*n* = 4/group), Cox-2-positive cells (*n* = 4/group), Bax-positive cells (*n* = 4/group), nitrotyrosine-positive cells (3–4 samples/group), and MDA-positive cells (*n* = 4/group) were counted.

### 2.6. CM-H_2_DCF-DA Assay

TDSCs were seeded at 40,000 cells in a 12-well plate and treated with 0.5 µg/mL of SeNPs and H_2_O_2_ (200 µM) in a serum-free medium. At 4 h after H_2_O_2_ exposure, the cells were washed and incubated with 1–10 µM of CM-H2DCF (Cat no.: C6827, Thermo Fisher Scientific, Waltham, MA, USA) for 30 min at 37 °C. The cells were rinsed, fixed with 4% paraformaldehyde, and counter-stained with DAPI. The fluorescence signals were viewed using a fluorescent microscope (Leica DM5500 B). The percentage of ROS-positive cells was counted. There were 3 samples/group.

### 2.7. Catalase Activity Assay

TDSCs were seeded at 40,000 cells in a 12-well plate in serum-free medium and the total protein was extracted. The collected protein was treated with 0.5 µg/mL of SeNPs and H_2_O_2_ (200 µM). At 2 h after H_2_O_2_ exposure, the catalase activity was measured using an assay kit (Cat no.: S0051, Beyotime Institute of Biotechnology, Suzhou, China). The optical density at 520 nm was measured. The activity level was normalized by the total protein content measured by the bicinchoninic acid (BCA) protein assay (Cat no.: #23227, Thermo Fisher Scientific, Waltham, MA, USA). There were 4 samples/group.

### 2.8. qRT-PCR

TDSCs were seeded at 40,000 cells in a 12-well plate and treated with 0.5 µg/mL of SeNPs and H_2_O_2_ (200 µM) in a serum-free medium for 4 h. qRT-PCR was performed using the QS7-Pro 384 and StepOne Plus 96 qPCR system (both from Applied Biosystems, Waltham, MA, USA). The relative gene expression was calculated using the 2^−ΔCT^ formula. We tested both *Gapdh* and *Actb* in the analysis of some markers and similar results were observed. Therefore, we used *Actb* as the house-keeping gene. The primer sequences are shown in Appendix A.

### 2.9. Statistical Analysis

Quantitative data are presented in boxplots. The sample sizes for the experiments in this study ranged from 3 to 5, which were considered small. In addition, the SeNPs without an H_2_O_2_ group were not included in this biological investigation because biologically, treatment with SeNPs is needed only when there is an increased production of ROS stimulated by H_2_O_2_ supplementation in this study, we therefore used the Kruskal–Wallis test to compare the difference in more than 2 groups. If there was a significant difference in the Kruskal–Wallis test, the Mann–Whitney U test was subsequently used to identify the differences. The effect size, r, was reported. A value of r at 0.1, 0.3, and 0.5 were regarded as small, medium, and large effects, respectively. All the data analyses were performed using the SPSS analysis software (SPSS Inc., Chicago, IL, USA, version 26.0). A value of *p* < 0.05 was regarded as statistically significant.

## 3. Results

### 3.1. Characterization of SeNPs

SeNPs used in this study were synthesized in our laboratory and the quality of the resulting particles was analyzed. Our results showed that the color of the reaction solution changed from colorless to red when selenous acid was added to PVA and ascorbic acid, indicating the synthesis of SeNPs from selenite oxyanion. XRD analysis showed a broad peak, suggesting that SeNPs were amorphous and lacked a good crystalline structure (Figure 1A). The transmission electronic microscopy (TEM) micrograph of dried SeNPs (Figure 1B) showed that the particle was spherical, and the selected area electron diffraction pattern (SAED) further demonstrated its amorphous nature. Energy dispersive X-ray analysis (EDX) was used to analyze the elemental composition of the selenium particles (Figure 1C). Selenium (Se), carbon (C), and oxygen (O) atoms were observed in the sample. The occurrence of carbon and oxygen atoms was due to the presence of residual polyvinyl alcohol (PVA) in the sample. The mean hydrated particle size and zeta potential of SeNPs were 91.3 nm (Figure 1D), and −7.6 mV, respectively, suggesting that SeNPs were in the nanometer scale. These SeNPs were used in the following experiments to examine the effects of selenium on the functions of H_2_O_2_-treated TDSCs.

### 3.2. Viability and Expression of Marker of Proliferation

H_2_O_2_ reduced the viability of TDSCs at 1 h and 4 h after treatment (both *p* < 0.01; *r* = 0.83) (Figure 2A). The addition of SeNPs significantly increased the viability of the H_2_O_2_-treated TDSCs (*p* < 0.05, *r* = 0.69 for SeNPs (0.25 µg/mL) treatment for 1 h; others (*p* < 0.01, *r* = 0.83) (Figure 2A). H_2_O_2_ reduced the percentage of Ki67^+^ cells (*p* < 0.05; *r* = 0.80) (Figure 2B), and the effect was reversed by co-treatment with SeNPs (*p* < 0.05; *r* = 0.80) (Figure 2B).

### 3.3. Oxidative Stress

H_2_O_2_ increased the percentage of ROS^+^ cells (*p* < 0.05; *r* = 0.80) and the addition of SeNPs at 0.5 µg/mL and 1 µg/mL returned the percentage of ROS^+^ cells to normal (both *p* < 0.05; *r* = 0.80) (Figure 3A). Similar results were observed in the mRNA expressions of selenoproteins (*Gpx1*, *Gpx3*, *Gpx4*, *Txnrd2*, *Selenom*), and enzymes implicated in oxidation (*Nox1*) and anti-oxidation (*Hmox1*, *Sod1*, *Cat*) (Figure 3B). H_2_O_2_ suppressed the expression of *Gpx3*, *Txnrd2*, and *Selenom* in TDSCs (all *p* < 0.05; *r* = 0.82) but had no effects on the expression of *Gpx1* and *Gpx4* at the concentration and time tested (both *p* > 0.05; *r* = 0.2 for *Gpx1* and *r* = 0.1 for *Gpx4*). SeNP supplementation significantly increased the expression of these five markers (all *p* < 0.05; *r* = 0.82) (Figure 3B). H_2_O_2_ elevated the expressions of oxidative enzyme *Nox1* (*p* < 0.05; *r* = 0.82) and anti-oxidative enzyme *Hmox1* (*p* < 0.05; r = 0.82), reduced the expression of Cat (*p* < 0.05; *r* = 0.82) but has no significant effect on the expression of *Sod1* (*p* > 0.05; *r* = 0.41) (Figure 3B). The addition of SeNPs significantly reversed the effects of H_2_O_2_ on the expression of *Nox1*, *Hmox1*, and *Cat* (all *p* < 0.05; *r* = 0.82) as well as increased the expression of *Sod1* (*p* < 0.05; *r* = 0.71) (Figure 3B). Similarly, SeNPs reduced the expressions of both protein oxidation products (nitrotyrosine) (*p* < 0.05; *r* = 0.8). (Figure 3C) and lipid peroxidation product (MDA) (*p* < 0.05; r = 0.82) (Figure 3D) in H_2_O_2_-treated cells. In addition, the catalase activity of H_2_O_2_-treated TDSCs increased after treatment with SeNPs (*p* < 0.05; *r* = 0.82) (Figure 3E).

### 3.4. Expression of Inflammatory Cytokines

H_2_O_2_ significantly upregulated the mRNA expressions of *Il6*, *Cox2*, and *Il1b* (all *p* < 0.05; *r* = 0.82 for *Il6* and *Cox2*; *r* = 0.87 for *Il1b*) (Figure 4A) as well as the percentages of IL-6^+^ cells (Figure 4B) and Cox-2^+^ cells (Figure 4C) (all *p* < 0.05; *r* = 0.82). SeNPs decreased the expressions of these inflammatory cytokines in TDSCs stimulated by H_2_O_2_ (all *p* < 0.05; *r* = 0.82) (Figure 4A–C).

### 3.5. Apoptosis

The exposure of TDSCs to H_2_O_2_ significantly elevated the mRNA expressions of pro-apoptotic and anti-apoptotic markers (all *p* < 0.05; *r* = 0.82) (Figure 5A) as well as the percentage of Bax^+^ cells (*p* < 0.05; *r* = 0.82) (Figure 5B) in TDSCs, as compared to the untreated group. Conversely, the addition of SeNPs significantly reduced the mRNA expressions of *Bax*, *Bad*, *Bcl2l1*, and *Bcl2* (all *p* < 0.05; *r* = 0.82) (Figure 5A) and the percentage of Bax^+^ cells (all *p* < 0.05; *r* = 0.82) (Figure 5B) triggered by H_2_O_2_. The mRNA expressions of *Bid* and *Casp3* also decreased after the addition of SeNPs but did not reach statistical significance (*p* > 0.05; *r* = 0.2) (Figure 5A).

### 3.6. Expression of Tenocyte Markers

The addition of H_2_O_2_ resulted in a significant decrease in the mRNA expressions of tenocyte markers (*Col1a1*, *Col3a1*, *Eln*, *Tnc* and *Dcn*) in TDSCs (all *p* < 0.01; *r* = 0.83) (Figure 6). The addition of SeNPs significantly increased the mRNA expressions of the tenocyte markers compared to the H_2_O_2_-stimulated group (*Dcn*: *p* < 0.05; *r* = 0.63; others: *p* < 0.01; *r* = 0.83) (Figure 6).

### 3.7. Expression of Transcription Factors Sirt1, FoxO1, and Nrf2

Treatment of TDSCs with H_2_O_2_ for 8 h significantly reduced the expressions of *Sirt1* and *Nrf2* (both *p* < 0.05; *r* = 0.82) but not *FoxO1* (*p* > 0.05; *r* = 0.41) (Figure 7). Supplementation of SeNPs significantly increased the expressions of *Sirt1* and *Nrf2* in TDSCs compared to the H_2_O_2_-treated group (both *p* < 0.05; *r* = 0.82 and 0.71 for *Sirt1* and *Nrf2*, respectively) (Figure 7).

## 4. Discussion

Our results showed that SeNPs increased the viability and expression of the marker of proliferation, as well as reduced the oxidative stress, inflammation, and apoptosis of H_2_O_2_-treated TDSCs. In addition, the expressions of tenocyte markers, *Sirt1* and *Nrf2* were significantly elevated in H_2_O_2_-treated TDSCs after SeNPs supplementation.

H_2_O_2_ is an important oxidative stress inducer. Overproduction of H_2_O_2_ causes oxidative damage, activation of endoplasmic reticulum stress, and apoptotic death of tenocytes [19]. In addition, H_2_O_2_ was also reported to impair colony formation, stemness marker expression, and differentiation capacity, as well as suppress cell migration, cell viability, apoptosis, and proliferation of TDSCs [20,21,22], similar to the results reported in this study. Strategies that can scavenge excessive ROS in the tendon may be useful for the promotion of tendon healing.

Previous studies have shown that inflammation promoted non-tenocyte differentiation of TDSCs, with higher expressions of osteogenic, chondrogenic, and adipogenic markers as well as reduced expression of tenocyte makers [6,7,8]. The pool of TDSCs available for tenogenic differentiation and regenerative repair was reduced, contributing to slow and poor tendon healing and occasionally ectopic bone formation at the late stage of tendon repair [23,24]. TDSCs are a major cell type involved in tendon repair after injury. The reduction in inflammation, oxidative stress, and apoptosis, and improvement in the survival and tenogenic activity of these cells after treatment with SeNPs in this study suggested that the delivery of SeNPs in a biodegradable biomaterial that supports its slow release might be used to promote tendon repair. This requires further research and evaluation of histology, and biomechanical properties of tendon after transplantation of SeNPs in a tendon injury animal model. SeNPs-loaded in biodegradable biomaterial can be prepared as an off-the-shelf, ready-to-use product for tendon repair. The quality control, storage conditions, and manufacturing costs are expected to be lower compared to the use of other biologics such as stem cell-based therapies. There is no standard treatment for reducing ROS production in tendons. Steroid and non-steroidal anti-inflammatory drugs (NSAIDs) are commonly used to reduce pain and inflammation after tendon injury. The inclusion of these drugs for comparison may provide more information about the relative efficacy of SeNPs in the promotion of tendon healing compared to the current practice, although there is no evidence of their long-term efficacy.

While there has been no study about the effects of SeNPs on TDSCs and tendon healing, our results were consistent with the effects of SeNPs reported in other tissues. Previous studies have shown that SeNPs alleviated homocysteine-mediated oxidation and endothelial dysfunction both in vitro and in vivo [25]. In addition to its anti-oxidative effect, SeNPs also exhibit anti-inflammatory and anti-apoptotic activities as well as modulate stromal cell differentiation. SeNPs inhibited the phosphorylation of nuclear factor of kappa light polypeptide gene enhancer in B-cells inhibitor alpha (IkB-α) and hence, prevented the release of nuclear factor kappa B (NF-κB), which is a central regulator of the synthesis of Cox-2, inducible nitric oxide synthase (iNOS), and many pro-inflammatory cytokines [26]. In addition, SeNPs also inhibited PGE_2_, which is an activator of pro-inflammatory cytokines like IL-6 and TNF-α [15]. In addition, selenium is also a critical regulator of macrophage activation. The addition of Se increased M1 to M2 polarization of interleukin-4 (IL-4)-treated macrophages [27]. Supplementation of SeNPs coated Ulva Lactuca polysaccharide (ULP-SeNPs) suppressed the activation and infiltration of macrophages in colon tissue by inhibiting the nuclear translocation of NF-κB [28]. In addition, SeNPs also increased the viability and osteogenic differentiation of human embryonic stem cell-derived mesenchymal stromal cells (MSCs) by modulating oxidative stress and preventing excessive ROS accumulation [29]. In addition, SeNPs reduced apoptosis and intracellular ROS levels as well as accelerated myogenic differentiation of C2C12 in the presence of H_2_O_2_ [16].

The absolute level of H_2_O_2_ in an injured tendon is not clear. The levels of H_2_O_2_ in tissues and plasma change rapidly, making it difficult to obtain an accurate result [30]. Additionally, the assays used can also affect the reliability of the results [30]. The levels of H_2_O_2_ in tissues and plasma vary depending on the physiological and pathological status. For instance, one study reported an increase in the level of H_2_O_2_ in a rat’s Achilles tendon after a contusion injury, although the absolute values were not provided [30]. In inflammatory diseases, the plasma H_2_O_2_ level was reported to increase from the normal range of 1–5 µM to 50 µM [31]. Furthermore, a study on elite soccer players found that serum oxidative stress, measured as H_2_O_2_ equivalent, was significantly higher in players with sonographic tendon alterations (1.75 ± 0.29 mM) compared to those without sonographic changes (1.3 ± 0.2 mM) [4]. This indicated that the oxidative stress in patients with tendon pathologies elevated, although the specific contribution of H_2_O_2_ in this case was not clear. Nonetheless, the dose of H_2_O_2_ (200 µM) used in this study was within the dose range used to induce oxidative stress in TDSCs in the literature (0.1 mM to 0.5 mM) [5,21,22,32]. Future studies should measure the absolute levels of H_2_O_2_ in tendon tissue after injury. Additionally, it would be beneficial to examine the in vitro effects of SeNPs across a range of H_2_O_2_ concentrations. This would offer valuable insights into the protective effects of SeNPs on the tendon under different oxidative stress microenvironments.

While all forms of selenium (i.e., inorganic selenium and organic selenium compounds) exhibit anti-oxidative and anti-inflammatory properties, they vary in their effectiveness in scavenging ROS and promoting tissue repair. In this context, Ng et al. [33] have discovered that 1,4-anhydro-4-seleno-D-talitol (SeTal) showed superior efficacy compared to D-selenomethionine (SeMet), D,L-trans-3.4-dihydroxyl-1-selenolane (DHSred), and 1,4-anhydro-D-talitol (Tal) in preventing pyrogallol-induced endothelial dysfunction in mouse aortas. SeNPs were chosen for this study due to their enhanced bioavailability, biological activity, and lower toxicity compared to inorganic or organic selenium. The small size and large surface area to volume ratio of SeNPs facilitate their absorption and cellular uptake, while mitigating cytotoxicity, as demonstrated in numerous studies [34,35]. Moreover, the incorporation of SeNPs in a biomaterial supports its controlled delivery to specific tissues or cells, optimizing its effects for tissue repair. Consequently, we chose to study the effects of SeNPs and did not compare the efficacy of nanoparticles and other forms of selenium on the functions of TDSCs under oxidative stress.

Selenium is an essential trace element in the diet. Potential side effects including reduction in body mass, changes in hepatotoxicity indices, and impairment of fatty acid, protein, lipid, and carbohydrate metabolism have been reported in laboratory animals [36]. The lowest adverse effect level (LOAEL) of SeNPs was reported to be 0.05 mg/kg. The no observed adverse effect level (NOAEL) was 0.22 mg/kg body weight per day and 0.33 mg/kg body weight per day for males and females, respectively [36]. Therefore, the doses of SeNPs that we tested in this study were safe.

While the mechanism of action of SeNPs on H_2_O_2_-treated TDSCs is unclear. A previous study has shown that selenium is taken up by cells and incorporated into the amino acid selenocysteine, which is vital for the biological functions of various selenoproteins [9]. Many selenoproteins act as strong antioxidants. The effects of SeNPs on TDSCs under oxidative stress are likely mediated by selenoproteins, as evidenced by the increased mRNA expression of glutathione peroxidase 1 (*Gpx1*), glutathione peroxidase 3 (*Gpx3*), glutathione peroxidase 4 (*Gpx4*), thioredoxin reductase 2 (*Txnrd2*), and selenoprotein M (*Selenom*) in this study. The selenoproteins break down H_2_O_2_. The improvement in redox balance in cells due to the functions of selenoproteins may modulate the expression of antioxidant enzymes including superoxide dismutase (*Sod1*) and catalase (*Cat*), as well as enhance catalase activity, as reported in this study.

The molecular mechanisms of the anti-apoptotic, anti-oxidative, and anti-inflammatory effects of SeNPs on TDSCs were not examined in this study. However, we reported the downregulation of *Casp3* after treatment of H_2_O_2_-exposed TDSCs with SeNPs. This suggested that SeNPs likely regulated cellular apoptosis by downregulating the expression of p53. This is consistent with the literature, which reported that intraperitoneal administration of selenium exhibited cardio-protective effects by downregulating *p53* and *Casp3* mRNA expression, as well as upregulating *Sirt1* mRNA expression [37]. FoxO1 and Nrf2 are known to regulate oxidative defense mechanisms, while Sirt1 was associated with the regulation of inflammation. SeNPs have been shown to exert their anti-oxidative and anti-inflammatory effects in various tissues by modulating the expressions of these transcription factors. Our results showed that H_2_O_2_ reduced the mRNA expressions of *Sirt1* and *Nrf2,* but not *FoxO1* at 8 h after treatment, and supplementation of SeNPs reversed the effects of H_2_O_2_ on the expressions of *Sirt1* and *Nrf2* in TDSCs. These findings were consistent with the literature reporting that selenium increased the expressions of Sirt1 and Nrf2, upregulated the expressions of anti-oxidative enzymes, and counteracted oxidative stress- and inflammation-induced tissue injury [5,38,39,40,41]. We did not observe an increase in the expression of *FoxO1* after SeNPs treatment in H_2_O_2_-stimulated TDSCs, despite the fact that SeNPs have been documented to activate the Wnt3a/β-catenin pathway, leading to the activation of FoxO1 and subsequently enhancing the expressions of catalase and SOD [42]. The expression of *FoxO1* was assessed only at a single time point. Further study should test the effect of SeNPs on the expression of *FoxO1* at different time points.

There has been no study about the effects of SeNPs on viability, oxidative stress, inflammation, apoptosis, and fate of TDSCs, making the work reported in this study original. However, this study is not without limitations. First, we did not examine the effects of SeNPs on tendon healing in animal models. This will be performed in our future experiments. Second, the targets of *Sirt1* and *Nrf2* genes in H_2_O_2_-exposed TDSCs after SeNPs treatment remain to be elucidated. Pharmacological inhibition or siRNA knockdown of Nrf2 and Sirt1 prior to SeNPs treatment of H_2_O_2_-stimulated TDSCs can help identify the potential downstream targets. This will be our future research direction.

## 5. Conclusions

In conclusion, SeNPs mitigated oxidative stress, inflammation, and apoptosis while enhancing the survival and expression of the marker of proliferation of TDSCs in an oxidative stress environment. Additionally, it promoted the fate of TDSCs towards the tenocyte lineage in the presence of such oxidative stress. The increased expressions of *Sirt1* and *Nrf2* likely mediated the anti-oxidative and anti-inflammatory effects of SeNPs. SeNPs hold promise as a novel intervention for promoting tendon healing.

## Figures and Tables

**Figure 1 antioxidants-13-01536-f001:**
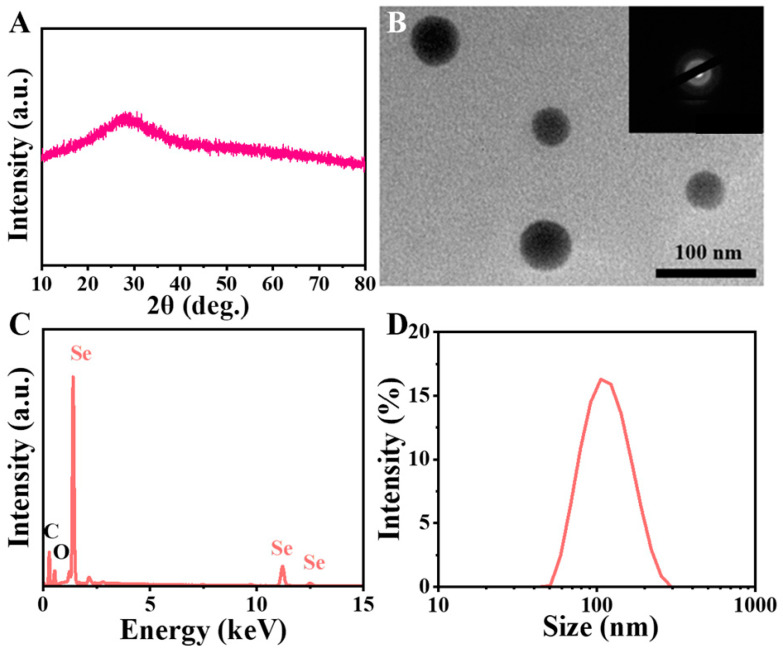
Characterization of SeNPs. (**A**) X-ray diffraction (XRD) pattern of the synthesized SeNPs; (**B**) transmission electron microscopy (TEM) bright field image and electron diffraction pattern of dried SeNPs; (**C**) energy dispersive X-ray analysis (EDX) spectrum of SeNPs; (**D**) the hydrated particle size of SeNPs.

**Figure 2 antioxidants-13-01536-f002:**
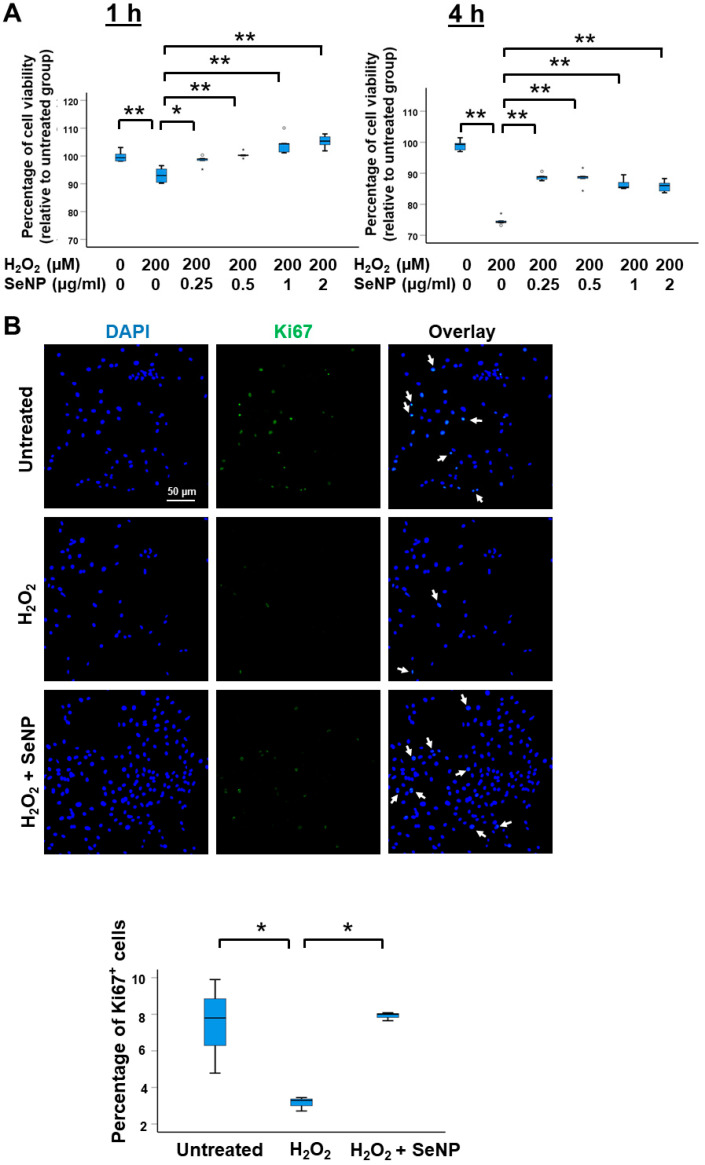
SeNPs increased the viability and expression of the marker of proliferation in H_2_O_2_-treated rat patellar TDSCs. (**A**) Percentage of cell viability of H_2_O_2_ (200 µM)-treated rat patellar TDSCs after treatment with SeNPs for 1 h and 4 h as measured by the AlamarBlue assay; *n* = 5/group; * *p* < 0.05; ** *p* < 0.01 show statistical significance between groups; “*” and “o” above or below the bar represent outliners and extreme values, of the dataset, respectively, and (**B**) IF staining of Ki67 and the percentage of Ki67^+^ cells of H_2_O_2_ (200 µM)-treated rat patellar TDSCs after SeNPs (0.5 µg/mL) supplementation for 4 h; scale = 50 µm; *n* = 3/group; * *p* < 0.05 for the indicated groups; white arrows: Ki67^+^ cells.

**Figure 3 antioxidants-13-01536-f003:**
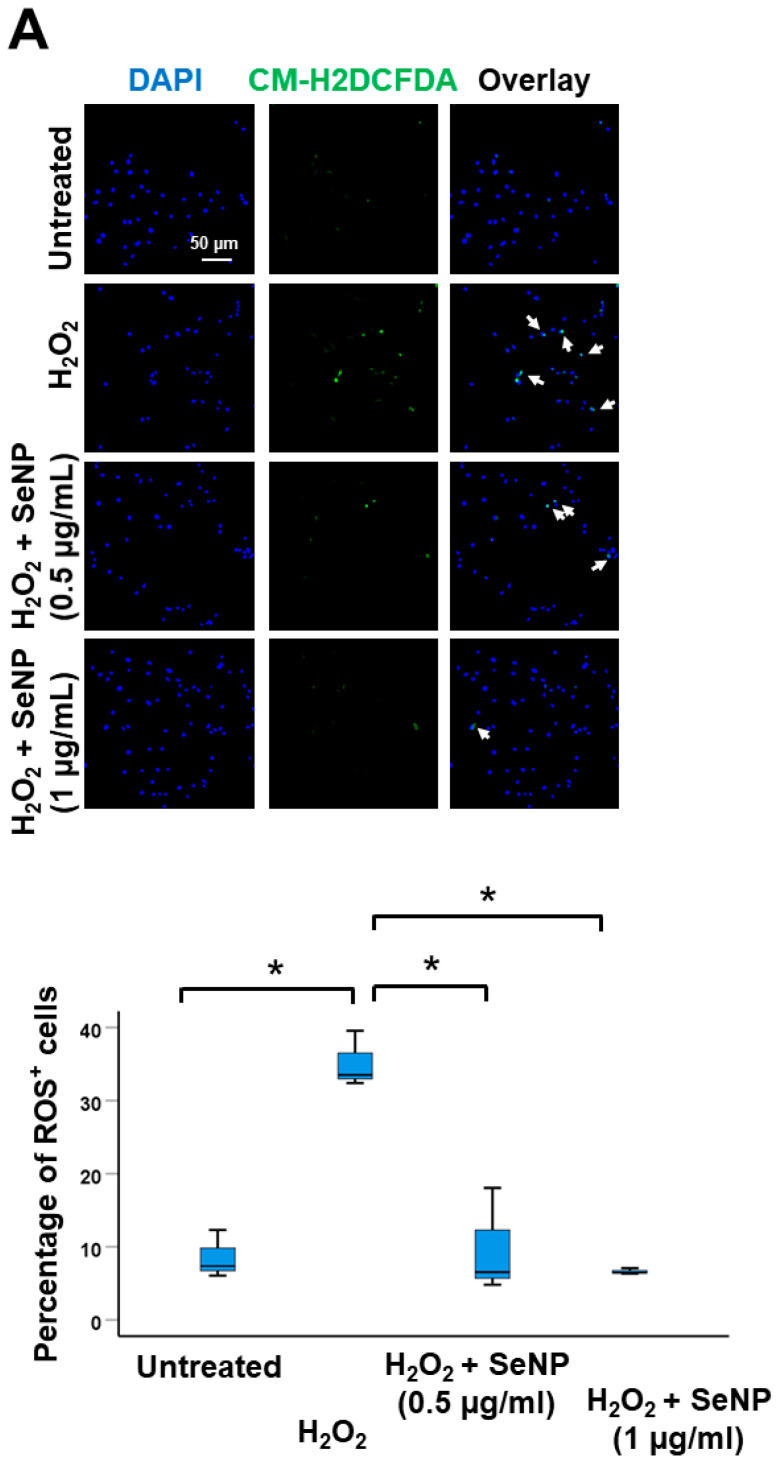
SeNPs reduced oxidative stress, as well as increased the expressions of selenoproteins, and the expressions and activities of anti-oxidative enzymes in H_2_O_2_-treated rat patellar TDSCs. (**A**) ROS staining and percentage of ROS^+^ cells of H_2_O_2_ (200 µM)-treated rat patellar TDSCs after treatment with SeNPs (0.5 µg/mL, 1 µg/mL) for 4 h as shown by CM-H2DCF-DA assay; scale = 50 µm; *n* = 3/group; * *p* < 0.05 shows statistical significance between groups; white arrows: ROS^+^ cells; (**B**) (from left to right, top to bottom) mRNA expressions of selenoproteins (*Gpx1*, *Gpx3*, *Gpx4*, *Txnrd2*, *Selenom*), oxidative enzyme (*Nox1*), and anti-oxidative enzymes (*Hmox1*, *Sod1*, *Cat*) of H_2_O_2_ (200 µM)-treated rat patellar TDSCs after treatment with SeNPs (0.5 µg/mL) for 4 h; *n* = 4/group; * *p* < 0.05 shows statistical significance between the groups; (**C**,**D**) IF staining of nitrotyrosine and MDA, and the percentages of nitrotyrosine^+^ cells and malonaldehyde^+^ cells after treatment with SeNPs (0.5 µg/mL) for 4 h in the presence of H_2_O_2_ (200 µM); scale = 50 µm; *n* = 3–4/group for nitrotyrosine and *n* = 4/group for malonaldehyde; * *p* < 0.05 shows statistical significance between groups; white arrows: nitrotyrosine^+^ cells or malonaldehyde^+^ cells; (**E**) the catalase activity (unit/mg) of H_2_O_2_ (200 µM)-stimulated rat patellar TDSCs after treatment with SeNPs (0.5 µg/mL) for 2 h as measured using an assay kit; *n* = 4/group; * *p* < 0.05 shows statistical significance between groups.

**Figure 4 antioxidants-13-01536-f004:**
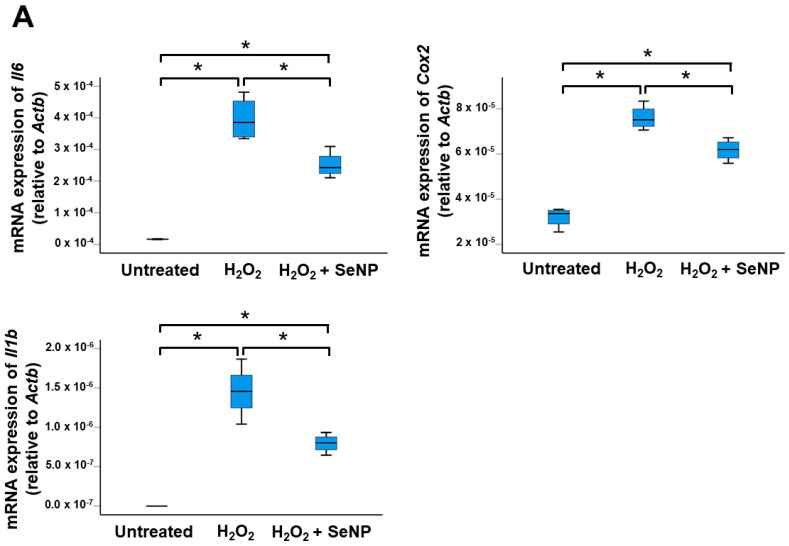
SeNPs suppressed H_2_O_2_-induced inflammation of rat patellar TDSCs. (**A**) (From left to right) mRNA expressions of inflammatory cytokines (*Il6*, *Cox2*, *Il1b*) in H_2_O_2_ (200 µM)-treated rat patellar TDSCs after treatment with SeNPs (0.5 µg/mL) for 4 h; *n* = 4/group; * *p* < 0.05 shows statistical significance between groups; (**B**,**C**) IF staining of IL-6 and Cox-2, and the percentages of IL-6^+^ cells and Cox-2^+^ cells after treatment with SeNPs (0.5 µg/mL) for 4 h in the presence of H_2_O_2_ (200 µM); scale = 50 µm; *n* = 4/group; * *p* < 0.05 shows statistical significance between groups; white arrows: IL-6^+^ cells or Cox-2^+^ cells.

**Figure 5 antioxidants-13-01536-f005:**
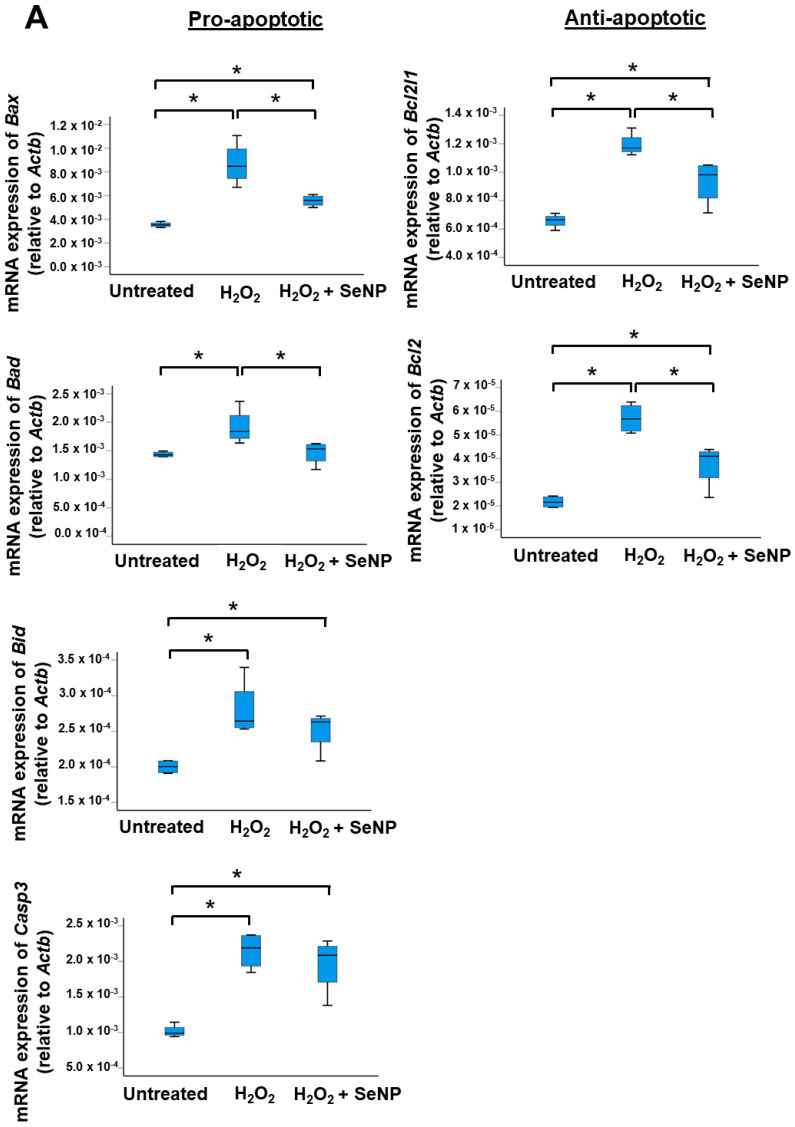
SeNPs suppressed H_2_O_2_-induced apoptosis of rat patellar TDSCs. (**A**) mRNA expressions of pro-apoptotic markers (*Bax*, *Bad*, *Bid*, *Casp3*) and anti-apoptotic markers (*Bcl2l1*, *Bcl2*) in H_2_O_2_ (200 µM)-treated rat patellar TDSCs after treatment with SeNPs (0.5 µg/mL) for 4 h; *n* = 4/group; * *p* < 0.05 shows statistical significance between groups; and (**B**) IF staining of Bax, and the percentage of Bax^+^ cells after treatment with SeNPs (0.5 µg/mL) for 4 h in the presence of H_2_O_2_ (200 µM); scale = 50 µm; *n* = 4/group; * *p* < 0.05 shows statistical significance between groups; white arrows: Bax^+^ cells.

**Figure 6 antioxidants-13-01536-f006:**
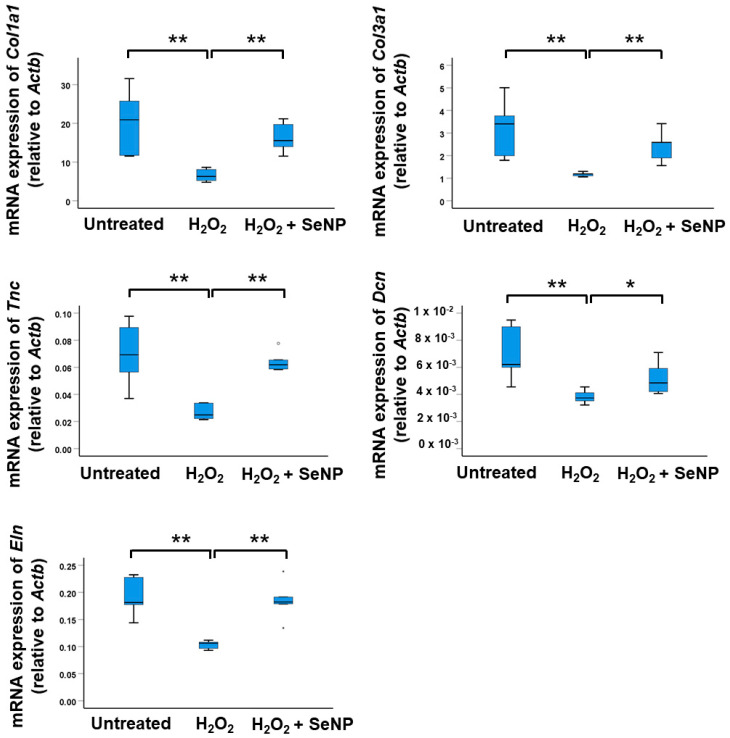
SeNPs rescued the reduced expressions of tenocyte markers in H_2_O_2_-treated rat patellar TDSCs. (From left to right, top to bottom) mRNA expressions of tenocyte markers (*Col1a1*, *Col3a1*, *Tnc*, *Dcn*, *Eln*) in H_2_O_2_ (200 µM)-treated rat patellar TDSCs after treatment with SeNPs (0.5 µg/mL) for 4 h; *n* = 5/group; * *p* < 0.05; ** *p* < 0.01 shows statistical significance between the groups; “*” and “o” above or below the bar represent outliners and extreme values, of the dataset, respectively.

**Figure 7 antioxidants-13-01536-f007:**
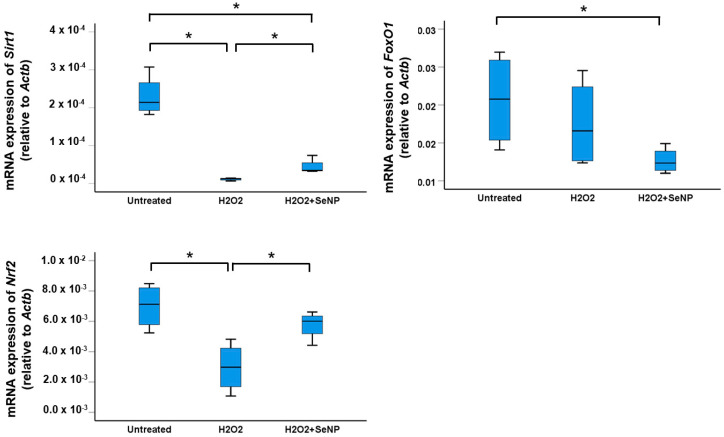
SeNPs upregulated the expressions of oxidative defense and anti-inflammation-related transcription factors in H_2_O_2_-treated rat patellar TDSCs. (From left to right, top to bottom) mRNA expressions of *Sirt1*, *FoxO1*, and *Nrf2* in H_2_O_2_ (200 µM)-treated rat patellar TDSCs after treatment with SeNPs (0.5 µg/mL) for 8 h; *n* = 4/group; * *p* < 0.05 shows statistical significance between groups.

## Data Availability

The raw data supporting the conclusions of this article are available on request as they are part of an ongoing study.

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
