# Peer review of "Selenium Nanoparticles Suppressed Oxidative Stress and Promoted Tenocyte Marker Expression in Tendon-Derived Stem/Progenitor Cells"

_antioxidants, 2024, doi:10.3390/antiox13121536_

Round 1

Reviewer 1 Report (Previous Reviewer 1)

antioxidants-3374530

This manuscript provides a revised version of the article “Selenium Nanoparticles Suppressed Oxidative Stress and Promoted Tenocyte Marker Expression in Tendon-Derived Stem/Progenitor Cells” (antioxidants-3172638). The manuscript has been improved considerably and my comments have been adequately addressed.

antioxidants-3374530

This manuscript provides a revised version of the article “Selenium Nanoparticles Suppressed Oxidative Stress and Promoted Tenocyte Marker Expression in Tendon-Derived Stem/Progenitor Cells” (antioxidants-3172638). The manuscript has been improved considerably and my comments have been adequately addressed.

Author Response

Reply: Thank you very much for your valuable comments in the first submission, which helped us improve the manuscript significantly.

Reviewer 2 Report (Previous Reviewer 2)

The authors have adequately answered all the reviewers' questions. 

There are no further questions, but the description added to the discussion is too long. For example, lines 506 to 519 on pages 26 to 27 could be described in one short sentence.

Author Response

The text of the axis titles and numerical values in the graphs is too small. Axis titles and other information should be explained in short words or abbreviations.

Reply: thank you very much for your comments. We have described the y-axis labels in the revised manuscript. We have also rearranged the graphs in Figure 7 to make the fonts bigger.

There are no further questions, but the description added to the discussion is too long. For example, lines 506 to 519 on pages 26 to 27 could be described in one short sentence.

Reply: Thank you very much for your comments. We have summarized the paragraph into one sentence.

Reviewer 3 Report (Previous Reviewer 3)

Review comments for “Selenium Nanoparticles Suppressed Oxidative Stress and Promoted Tenocyte Marker Expression in Tendon-Derived Stem /Progenitor Cells”.

In this manuscript, the authors investigated the effects of selenium nanoparticles (SeNP) on tendon-derived stem/progenitor cells (TDSCs) under oxidative stress conditions. The authors demonstrated that SeNP treatment enhanced TDSC viability and proliferation while reducing oxidative stress, inflammation, and apoptosis in H2O2-treated TDSCs. Notably, SeNP promoted tenocyte marker expression and activated the Sirt1 and Nrf2 pathways, suggesting its potential role in tendon healing. The study provides novel insights into SeNP's protective mechanisms against oxidative stress in tendon tissue and presents a promising therapeutic approach for promoting tendon repair.

Here are comments for the manuscript.

Major Strengths:

1. The experimental design is well-structured with appropriate controls and methodology

2. The mechanistic insights into SeNP's effects through Sirt1 and Nrf2 pathways are valuable

3. The data presentation and statistical analysis are robust

Minor suggestions for revision:

1. Mechanistic validation of Nrf2/Sirt1 pathway using inhibitors or siRNA knockdown greatly enhances the quality of this manuscript.

2. Consider adding a brief discussion on the potential clinical translation challenges

3. A graphical abstract might help readers better understand the key findings

4. Methods section: include supplier information for key reagents

Review comments for “Selenium Nanoparticles Suppressed Oxidative Stress and Promoted Tenocyte Marker Expression in Tendon-Derived Stem /Progenitor Cells”.

In this manuscript, the authors investigated the effects of selenium nanoparticles (SeNP) on tendon-derived stem/progenitor cells (TDSCs) under oxidative stress conditions. The authors demonstrated that SeNP treatment enhanced TDSC viability and proliferation while reducing oxidative stress, inflammation, and apoptosis in H2O2-treated TDSCs. Notably, SeNP promoted tenocyte marker expression and activated the Sirt1 and Nrf2 pathways, suggesting its potential role in tendon healing. The study provides novel insights into SeNP's protective mechanisms against oxidative stress in tendon tissue and presents a promising therapeutic approach for promoting tendon repair.

Here are comments for the manuscript.

Major Strengths:

1. The experimental design is well-structured with appropriate controls and methodology

2. The mechanistic insights into SeNP's effects through Sirt1 and Nrf2 pathways are valuable

3. The data presentation and statistical analysis are robust

Minor suggestions for revision:

1. Mechanistic validation of Nrf2/Sirt1 pathway using inhibitors or siRNA knockdown greatly enhances the quality of this manuscript.

2. Consider adding a brief discussion on the potential clinical translation challenges

3. A graphical abstract might help readers better understand the key findings

4. Methods section: include supplier information for key reagents

Author Response

  1. Mechanistic validation of Nrf2/Sirt1 pathway using inhibitors or siRNA knockdown greatly enhances the quality of this manuscript.

Reply: Thank you very much for your suggestions. This experiment is in our future plan, which we will explore the downstream targets of Nrf2/Sirt1 at the same time. We have added this as a limitation for the current study. 

  1. Consider adding a brief discussion on the potential clinical translation challenges

Reply: We expected that clinical translation would be easier compared to biologics. SeNP-loaded in biodegradable biomaterial can be prepared as an off-the-shelf, ready-to-use product for tendon repair. The quality control, storage conditions, and manufacturing costs are expected to be lower compared to the use of other biologics such as stem cell-based therapies. This information are added in the revised manuscript.

  1. A graphical abstract might help readers better understand the key findings

Reply: A graphical abstract is added in the revised manuscript.

  1. Methods section: include supplier information for key reagents

Reply: The supplier information of key reagents are included in the revised manuscript.

This manuscript is a resubmission of an earlier submission. The following is a list of the peer review reports and author responses from that submission.

Round 1

Reviewer 1 Report

antioxidants-3172638

Selenium Nanoparticles Suppressed Oxidative Stress and Promoted Tenocyte Marker Expression in Tendon-Derived Stem/Progenitor Cells

This article describes the effect of selenium nanoparticles (SeNP) on H2O2-challenged rat tendon-derived stem/progenitor cells (TDSC). H2O2 affected TDSC viability and proliferation, enhanced oxidative stress, inflammation, and apoptosis, and reduced tenocyte marker expression. The application of SeNP, however, had the opposite effect and reversed the detrimental effect of H2O2. The authors conclude that SeNP may represent a promising approach for improving tendon healing.

The study is well designed, carried out properly, and technically sound. The manuscript is well written and the presentation of the data is straightforward and clear. The results are comprehensible and conclusive. Thus, there are only some minor points that should be addressed.

1. Introduction: Is anything known about the molecular mechanism(s) with which selenium (either in selenoproteins or SeNP) mediates its antioxidant/antiapoptotic effects?

2. All abbreviations should be defined in the text.

3. Materials and Methods: The number of cells used for each experiment has to be included.

4. Section “Study design”: The authors used 200 µM H2O2 for stimulation. Please comment on the relation of this concentration to the H2O2 concentration in tissue.

5. Section “TDSC isolation” (“… TDSCs at early passages were used …”): The maximum passage to which the cells were used should be mentioned.

6. Results: When describing the results of the experiments, please also include the effect sizes.

7. Section 3.2 (Ki67 staining): Did the authors observe an increased number of SeNP-treated cells over time, i.e., beyond 4h? If not, I suggest to use the term "marker of proliferation" instead of "proliferation".

8. Figures 2 and 6: Please replace the “*” for the indication of outliners by another symbol to avoid confusion with the indication of significance.

9. Figures 3 and 5: Please increase size and resolution.

10. Line 223 (“… treatment with SeNP (0.5 μg/mL) for 4 h …”): Please add 1 μg/mL as the second SeNP concentration used.

11. Line 251 (“… SeNP significantly inhibited the mRNA expression …”): As there is no complete inhibition of Bax, Bcl2l1 and Bcl2, I suggest to use the term “reduced”.

12. Supplement: Please complete the information on Dcn and Actb in Table S1.

Reviewer 2 Report

This manuscript describes that in an experimental TDSCs culture selenium microparticles are protective against the adverse effects of oxidative stress. The authors showed that hydrogen peroxide (0.2 mM) inhibited the viability and proliferation of TDSCs by 1-4 hours with increased ROS production, which was significantly inhibited when selenium particles (SeNP) were added. Also, hydrogen peroxide-induced (inflammatory cytokines, apoptosis-related, and oxidative stress-inducing enzymes) or suppressed (tendon-specific markers) gene expression was reversed by SeNP. Thus, the authors argue that selenium can be a protective therapeutic modifier for inhibited healing of injured tendons. Although this simple experimental study showed clinically hopeful findings, its novelty is not necessarily high, since the adverse effects of hydrogen peroxide on tendon cell culture and the anti-inflammatory effects of selenium in other cells have already been known. Specific comments are as follows.

Major Points

1.     In the abstract, the authors state that they examined the mechanisms of selenium effects on TDSCs treated with hydrogen peroxide but actually haven’t examined any mechanism of action. It may not be necessary because the mechanism of action is inherently already elucidated (The abstract lacks mention of why the authors used selenium in this experiment; antioxidant effect). The authors added SeNP as an antioxidant molecule to prevent ROS production induced by hydrogen peroxide.

2.     in the above sense, examining the differences in the expression levels of various genes is more of a confirmation of the results and does not bring us closer to elucidating the mechanism. They showed that the expression of various genes was affected by oxidative stress, and was protected by SeNP, but we are not necessary to examine the mechanism for it. Rather than examining the mechanism, the authors just confirmed in various ways that selenium exerted antioxidant activity and reduced the adverse effects of hydrogen peroxide on cells.

3.     although 3 groups (control, hydrogen peroxide, and hydrogen peroxide plus SeNP) were compared, 2x2=4 groups (hydrogen peroxide (+ or -) and SeNP added (+ or -)) should be used for the comparison. It seems that a 2-way ANOVA should be used instead of a Kruskal-Wallis analysis. It is not stated whether the normality of the values was tested or not.

Minor Points

1.     Specify how hydrogen peroxide was added. Please specify whether or not the hydrogen peroxide was washed out. Also, please specify when and how the selenium particles were added.

2.     The description in the Results section does not have a descriptive flow. For example, the first step is to check the quality of the SeNPs, but the description starts abruptly. Please start with a brief explanation of the meaning of this description. The text mentions “XRD analysis,” but that term is not used in the Materials and Methods section.

3.     On page 4, lines 175 and 176, is “Se, C and O” referring to selenium, carbon and oxygen atoms?

4.     Most of the discussion section is just an introduction and a repetition of the results. Please remove it to avoid duplication.

5.     The reported study is simple and does not require a lengthy discussion, but if the goal is to use selenium as a healing accelerator for the treatment of injured tendons, please explain in the Discussion how you plan to achieve this. Please also describe the in vivo animal experiments you plan to conduct before practical use. Please describe possible side effects and measures to prevent them. If there are other candidate drugs other than selenium that are expected to have similar effects, please describe them. If you think selenium is the most effective, please describe why.

6.     Se” in page 9, lines 298 and 299 should be written as ‘Selenium’.

Reviewer 3 Report

Review comments for antioxidants-3172638.

In this manuscript, the authors examined the therapeutic effects of selenium nanoparticles (SeNP) on tendon-derived 19 stem/progenitor cells. They claimed that SeNP attenuated H2O2-mediated cytotoxicity. Next, they also examined inflammatory cytokine expression and found that SeNP inhibited H2O2-mediated inflammatory cytokine expression. Finally, they examined tenocyto markers and found that SeNP rescued H2O2-mediated reduction.

This manuscript is of interest from the point of possibility of SeNP as therapeutic medicine for tendon injury. However, there are some concerns, and they are discussed below.

1. Comparison of SeNP and Selenous acid in each response.

The authors indicated fantastic protective effect of SeNP, however, the reviewer is curious these protective effects are dependent to nano particle or selenium itself. Comparisons are necessary to clarify it.

2. Mechanism how SeNP works.

As the authors exhibited several protective effects of SeNP, and augmentation of anti-oxidation, the reviewer is curious why SeNP augments anti-oxidation. Transcription factors such as Nrf2, Sirtuin, and FOXOs are reported to regulates anti-oxidative enzymes, therefore, the reviewer recommends to explore the changes in the expression of these transcription factors.

3. No animal experiment

Though the authors used experimental rats for the source of primary tendon cells, there is no in vivo experiments which revealing the effects of SeNP on the recovery of tendon injury. These animal experiments would enforce the authors results extensively.

Review comments for antioxidants-3172638.

In this manuscript, the authors examined the therapeutic effects of selenium nanoparticles (SeNP) on tendon-derived 19 stem/progenitor cells. They claimed that SeNP attenuated H2O2-mediated cytotoxicity. Next, they also examined inflammatory cytokine expression and found that SeNP inhibited H2O2-mediated inflammatory cytokine expression. Finally, they examined tenocyto markers and found that SeNP rescued H2O2-mediated reduction.

This manuscript is of interest from the point of possibility of SeNP as therapeutic medicine for tendon injury. However, there are some concerns, and they are discussed below.

1. Comparison of SeNP and Selenous acid in each response.

The authors indicated fantastic protective effect of SeNP, however, the reviewer is curious these protective effects are dependent to nano particle or selenium itself. Comparisons are necessary to clarify it.

2. Mechanism how SeNP works.

As the authors exhibited several protective effects of SeNP, and augmentation of anti-oxidation, the reviewer is curious why SeNP augments anti-oxidation. Transcription factors such as Nrf2, Sirtuin, and FOXOs are reported to regulates anti-oxidative enzymes, therefore, the reviewer recommends to explore the changes in the expression of these transcription factors.

3. No animal experiment

Though the authors used experimental rats for the source of primary tendon cells, there is no in vivo experiments which revealing the effects of SeNP on the recovery of tendon injury. These animal experiments would enforce the authors results extensively.